# Broad Specific Xyloglucan:Xyloglucosyl Transferases Are Formidable Players in the Re-Modelling of Plant Cell Wall Structures

**DOI:** 10.3390/ijms23031656

**Published:** 2022-01-31

**Authors:** Maria Hrmova, Barbora Stratilová, Eva Stratilová

**Affiliations:** 1Jiangsu Collaborative Innovation Centre for Regional Modern Agriculture and Environmental Protection, School of Life Science, Huaiyin Normal University, Huai’an 223300, China; 2School of Agriculture, Food and Wine & Waite Research Institute, University of Adelaide, Glen Osmond, SA 5064, Australia; 3Institute of Chemistry, Centre for Glycomics, Slovak Academy of Sciences, SK-84538 Bratislava, Slovakia; barbora.stratilova@savba.sk (B.S.); eva.stratilova@savba.sk (E.S.); 4Faculty of Natural Sciences, Department of Physical and Theoretical Chemistry, Comenius University, SK-84215 Bratislava, Slovakia

**Keywords:** crystal structures, evolutionary history, glycoside hydrolase family 16, mechanism of catalysis, molecular modelling and dynamics, transglycosylation reactions, substrate binding

## Abstract

Plant xyloglucan:xyloglucosyl transferases, known as xyloglucan endo-transglycosylases (XETs) are the key players that underlie plant cell wall dynamics and mechanics. These fundamental roles are central for the assembly and modifications of cell walls during embryogenesis, vegetative and reproductive growth, and adaptations to living environments under biotic and abiotic (environmental) stresses. XET enzymes (EC 2.4.1.207) have the β-sandwich architecture and the β-jelly-roll topology, and are classified in the glycoside hydrolase family 16 based on their evolutionary history. XET enzymes catalyse transglycosylation reactions with xyloglucan (XG)-derived and other than XG-derived donors and acceptors, and this poly-specificity originates from the structural plasticity and evolutionary diversification that has evolved through expansion and duplication. In phyletic groups, XETs form the gene families that are differentially expressed in organs and tissues in time- and space-dependent manners, and in response to environmental conditions. Here, we examine higher plant XET enzymes and dissect how their exclusively carbohydrate-linked transglycosylation catalytic function inter-connects complex plant cell wall components. Further, we discuss progress in technologies that advance the knowledge of plant cell walls and how this knowledge defines the roles of XETs. We construe that the broad specificity of the plant XETs underscores their roles in continuous cell wall restructuring and re-modelling.

## 1. Introduction

This review is presented in the three inter-connected sections that inform on: plant cell walls (CW) as multi-composite hydrogels, including their composition, structure and organisation (Section 2) and plant xyloglucan:xyloglucosyl transferases and their nomenclature, classification, catalytic mechanisms, structural properties and substrate specificity with xyloglucan (XG) and other than XG-derived donor and acceptor substrates (Section 3). In the last Section 4, we focus on plant CW modifications and re-modelling, specifically on the function of xyloglucan:xyloglucosyl transferases in CW re-modelling and dynamics, notwithstanding methods of investigation. We also explore recent progress made in technologies that advance the knowledge of CWs and how this knowledge defines the function of these transferases. The additional goal of this review is to examine the roles of the xyloglucan:xyloglucosyl transferase family in plant CWs and environmental contexts, and not as isolated catalytic entities. We expect that the detailed definition of catalytic properties of these enzymes will contribute to the clarification of the biological function of xyloglucan:xyloglucosyl transferases, in the framework of the recent models of primary and secondary plant CW architectures.

## 2. Plant Cell Walls Are Multi-Composite Hydrogels

*(a) Composition of plant CWs—*Plant CWs are multi-composite hydrogels that are predominantly comprised of polysaccharides linked by covalent and non-covalent linkages, which collectively define the properties of CWs [1,2,3]. Other important constituents of plant CWs are proteins including no-catalytic expansins, and organic and inorganic compounds, including non-fermentable polyphenolic lignins. The properties of plant CWs crucially depend on the characteristics of their building blocks, chemical linkages of components, and spatial disposition. Plant CWs have certain common characteristics regardless of their phylogenetic origin, and variations exist between species, organs, tissues, developmental stages during embryogenesis, vegetative and reproductive growth, adaptations to living environments, and biotic and abiotic stresses [4].

Plants form two major types of CWs, termed (i) primary and (ii) secondary CWs. These two major types reflect the chemical composition and structure that fulfil fundamental roles in mono- and di-cotyledonous species. Primary CWs typically contain the networks of cellulose, made of (1,4)-β-d-glucan-linked chains forming micro-fibrils tightly bound via hydrogen bonds. These supra-molecular cellulosic structures form the backbone of primary CWs. Cellulose, as the most abundant polymer, is found in CWs of higher plants, red, brown and green algae, and other species such as *Oomycetes*, *Ameobozoa* and *Cyanobacteria* [4]. Cellulose micro-fibrils are tethered by cross-linking hemicelluloses [5] and embedded in most dicotyledonous plants in the matrix of pectins or glycoproteins, which through hydrogen bonds and other interactions provide the frameworks to activate the CW assembly [6,7]. These hemicelluloses include XGs, which are the major form of cross-linking glycans in dicotyledonous plants, while xylans (arabinoxylans, glucuronoarabinoxylans), mannans (galactomannans, glucomannans, galactoglucomannans) and (1,3;1,4)-β-d-glucans (mixed linkage or MLG glucans) substitute XGs in monocotyledonous and lower plants [8]. Pectins composed of the homogalacturonan main chains could be calcium-cross-linked [9], while branched pectin regions are made of rhamnogalacturonan I and rhamnogalacturonan II [10,11].

Conversely, secondary CWs that are characteristic of woody plants and vascular tissues after their growth ceases, and are more rigid in comparison to the primary CWs, as they contain fewer pectins, and are reinforced by hydrophobic phenylpropanoid lignin polymers [12,13,14]. Wood formation occurs via radial proliferation of vascular cambium cells and the deposition of secondary CW layers [15]. This is critical for the mechanical stability of woody plants as it affects the chemical, mechanical and elastic properties of wood and ligno-cellulosic materials produced from wood [16].

As defined, cellulose micro-fibrils are tethered by the cross-linking hemicelluloses, such as XGs that are the most abundant hemicelluloses in the primary CWs of dicotyledonous plants. The biosynthesis of XGs [17] including green algae [18] was described in plants, while XGs were not found in red and brown algae [4], which is also accompanied by the lack of XG-modifying enzymes [19]. XG forms the chains of repeating (1,4)-β-d-linked glucopyranosyl residues, with the C-6 carbon carrying α-d-xylopyranosyl residues [20], which could be substituted by galactopyranosyl residues on the C-2 carbons (β-d-Galp-(1,2)-α-d-Xylp) and the galactosyl residues that on the C-carbons carry fucopyranosyl substitutions (α-l-Fucp-(1,2)-β-d-Galp-(1,2)-α-d-Xylp) (Figure 1), although, in moss and liverwort XGs, arabinopyranosyl [21] and galacturonate [22] substituents are found instead the fucopyranosyl residues. This suggests that the structure of XGs differs amongst plants, organs and tissues even of the same plant [23]. The XG backbone is synthesised, using UDP-glucose, by XG:glucan synthases encoded by the C subfamily of cellulose synthases-like genes [17,24], implying that the activity of an additional enzyme using UDP-xylose, namely an XG:xylosyltransferase is needed to generate XGs [25,26]. d-Xyl residues are additionally modified with d-Gal, and this decoration is mediated by XG:galactosyltransferases, and d-Gal moieties could in some XGs carry l-fucosyl moieties, which is catalysed by XG:fucosyltransferases [26,27]. Enzymes in each XG biosynthetic step occur in multiple isoforms with a discrete substrate specificity, as it was described for the XG:xylosyltransferase from *Arabidopsis thaliana* [26].

*(b) Plant cell wall structure and organisation*—The chemical and structural parallels between cellulose and XGs underlie the conformational homology, which results in noncovalent interactions [27]. The networks of cellulose and XGs support the structure of primary CWs, which underlines their flexibility and strength [28,29]. The most recognised primary CW model of dicotyledonous plants [30] is based on the linear micro-fibrils of cellulose, each consisting of 32 micro-fibrils associated via hydrogen bonds, where para-crystalline cellulose micro-fibrils are interwoven and bridged by XGs. Consistent with this model is the function of pectin, which is attributed to the function of a gelling milieu that permeates the space between cellulosic and XG polymers.

Microscopic techniques allowed the development of sophisticated CW models that are based on cellulose micro-fibrils, composed of eight or sixteen associating cellulose sub-units [31]. These hot spot micro-fibrils are linked with each other by a thin layer of XG polysaccharides [32], where pectins fulfil the role of the filling materials between cellulosic and XG polymers. Various pectins that make close contacts with both cellulose and XGs, ensure the flexibility but also the strength of CWs [33,34]. The processes of these hot spots formations are unknown, but it is assumed that they could be formed ad hoc or with the participation of enzymes [34].

While sufficient information is available on the role of XGs in CW modifications during plant growth, no information is accessible on the participation of XGs in CW formation. Recent work with the *xxt1 xxt2* mutant lacking XGs suggested that the formation of cellulose networks could be XG-independent [35]. As for the necessity of XG for CW regeneration in plant protoplasts, it was established that XG is non-essential for the formation of the cellulose networks and that the cellulose networks formed in the absence of XGs provide sufficient tensile strength to the primary CW in *Arabidopsis* protoplasts [31].

As alluded to above, CWs are diverse, which underscores the function of cells, and this could be directly linked to the elusive structural changes of polymeric components, their quantity, ratios, and underpins their interactions. The syntheses of CW polysaccharides occur due to the cooperative activities of various glycosyl transferases (GTs) or synthases [17,36,37,38], operating usually in the Golgi apparatus and associated vesicles [39]. These enzymes are transported to CWs by secretory vesicles, but other enzymes, such as cellulose synthases [40,41] and callose synthases [42] locate directly to the plasma membranes [43], meaning that their polysaccharide products are exported to CWs, although the latter structural polysaccharides are often heterogeneous. It is assumed that a wide range of activated sugar donors required for the GTs activities in isomeric forms could give rise to a variety of glycosidic linkages.

As the life processes in plants are guided by CW organisation, involving elongation, expansion and loosening, in addition to GTs and synthases, these processes are also governed by non-catalytic expansins [44,45], and enzymes with hydrolytic, esterase and lyase (non-hydrolytic addition or removal of groups from substrates) activities. There is a large number of these enzymes that could potentially modify the structural polysaccharides in situ by breaking linkages, esterifying or de-esterifying saccharides [46], and incorporating new saccharide components in CWs or re-constructing polymers through cross-linking. The latter processes are mediated by xyloglucan:xyloglucosyl transferases also known as xyloglucan endo-transglycosylases (XETs) [47,48], whereby these enzymes are one of the key glycosidic bond-formation biocatalysts taking part in plant CW expansion, modification, reconstruction and re-modelling.

## 3. Plant Xyloglucan:Xyloglucosyl Transferases

Xyloglucan:xyloglucosyl transferases or xyloglucan endo-transglycosylases (XET), classified under EC 2.4.1.207, transfer the glycosyl groups from one glycoside to another. These enzymes were discovered in 1992 independently in bean epicotyls [49], nasturtium seeds [50], and pea, tomato and other plant extracts [51], and since their discovery, significant knowledge has been accumulated on their mode of action.

*(a) Nomenclature**and classification*—The nomenclature of XET enzymes is defined by the International Union of Biochemistry and Molecular Biology (IUBMB) and International Union of Pure and Applied Chemistry Biochemical Nomenclature Committee [52], the Kyoto Encyclopaedia of Genes and Genomes Enzyme Database, and the BRENDA Comprehensive Enzyme System [53]. These nomenclatures either consider the transfer of the ‘glycosyl’ (xyloglucan endo-transglycosylase) [49,50,51] or ‘glucosyl’ (xyloglucan endo-transglucosylase) [54,55] groups.

The complex view on XET enzyme nomenclature and classification is provided by the Carbohydrate-Active enZYmes Database (CAZy) [56] and CAZy*pedia* [57], which classify entries based on protein tertiary structures, substrate specificities and evolutionary history or phylogenomic relationships. XETs are listed in the glycoside hydrolase (GH) family 16, and not in a glycoside transferase (GT) Leloir group of enzymes. According to the Enzyme Commission and International Union of Biochemistry and Molecular Biology, the latter group involves enzymes that use activated sugars (for example UDP-glucose) as the glycosyl donors, and transfer the glycosyl groups to the nucleophilic glycoside (or other) acceptors. The GH16 family, according to tertiary structural features is sub-divided into 23 subfamilies [58], where the subfamily GH16_20 includes XETs but also xyloglucan endohydrolases (XEHs, EC 3.2.1.151) [59]; the latter group catalyses predominantly hydrolytic reactions of XGs. Thus, the GH16_20 group contains the products of *XTH* (xyloglucan transglycosylase/hydrolase) genes encoding both types of XG-modifying enzymes which notably display close similarities in tertiary structures [60].

*(b) Catalytic mechanism*—The catalytic function of XETs is defined by breaking a glycosidic bond between 1,4-β-d-linked glucosyl residues of XGs (or other polysaccharides) and transferring the xyloglucanyl or another glycoside moiety onto the O-4 atom of a non-reducing end of a glycoside acceptor; this acceptor could be XG, XG-oligosaccharide or another glycoside. This constitutes a so-called ping-pong bi-bi reaction mechanism and not a sequential one [61,62]. Recently, the definition of XET substrate specificity has significantly expanded, and thus the previous information on XETs has become obsolete [47,48,63,64,65,66,67,68,69].

The first steps of the transglycosylation and hydrolytic reactions catalysed by XTHs (that is XETs and XEHs) are binding and cleavage of donor glycoside substrates. The difference between XETs and XEHs occurs in the second step, in which the fragment with the original non-reducing end of the donor substrate is transferred to an acceptor, which in the case of a typical transglycosylase is a saccharide, while in the case of a hydrolase, it is an activated water molecule [56,70]. This second step had the key importance for the nomenclature and classification of XETs as transferases.

The transglycosylation mechanism catalysed by XET enzymes proceeds in two stages with two transition states. The first step involves the deprotonation of a carboxylate that operates as a nucleophile and attacks the anomeric carbon of the donor substrate, forming the glycosyl–enzyme intermediate complex, with the participation of an acid/base carboxylate. In the hybrid aspen *Populus tremulus x tremuloides* PttXET16A, the nucleophile attacking the anomeric carbon is Glu85, while Glu89 acts as an acid/base (has a dual role), that is it protonates the aglycon saccharide and releases it, and subsequently, it de-protonates the glycoside acceptor. The third important residue in PttXET16A is Asp87 positioned between Glu85 and Glu89 (signature motif ExDxE), which forms a tight bond with the Glu85 nucleophile, and interacts with the substrates through hydrogen bonds during substrate binding. The nucleophile must be de-protonated for the donor substrate attack, while Asp87 and Glu89 remain protonated. The preference for the transfer reaction instead of hydrolytic one by XETs leads to the re-ligation of the nascent donor end of one saccharide to the non-reducing end of the acceptor substrate [60,71]. Curiously, in the crystal structure of PttXET16A (‘true’ XET without hydrolytic activity) [71], only a few water molecules were resolved near the donor and acceptor substrate binding sites, while significantly more water molecules were located around the corresponding sites in XEHs.

Typically, hydrolases also transglycosylate under high(er) substrate concentrations [70,72], when in the later stages of reactions, the glycosyl–enzyme intermediate complex interacts with another sugar molecule (instead of an activated water molecule), thus shifting the reaction equilibrium towards the transfer products. Contrary to these high-substrate-induced transglycosylation reactions, ‘true’ transglycosylases such as XETs encounter a second saccharide regardless of a substrate concentration. While transglycosylation reactions catalysed by hydrolases occur under high(er) concentrations of substrates and generated products, this is not the case of XETs.

In addition to XETs, other plant enzymes with transglycosylase activities were reported, often without providing protein sequence information, with trivial descriptions such as mannan endotransglycosylase/hydrolases recognising (1,4)-β-d-mannan-derived polysaccharides [73], xylan endo-transglycosylases functionalising heteroxylan polysaccharides [74,75,76], mixed-linkage glucan:xyloglucan endotransglucosylases acting on (1,3;1,4)-β-d-glucans [77,78], and hetero-trans-β-glucanases (HTGs) functionalising cellulose [48,79]. However, terming these enzymes as such disguises a fact that some of them could in fact be non-specific XETs.

*(c) Structural properties, evolutionary relationships, enzyme activity methods*—The crystal structures of the XET enzyme PttXET16A (Protein Data Bank—PDB accessions and 1UMZ and 1UN1 at respective 1.80 Å and 2.10 Å resolution; 1UMZ in complex with the XLLG acceptor) [60,71,80] offers insights into its atomic architecture and catalytic mechanism. The crystal structure revealed that PttXET16A has the β-jelly-roll topology and folds into a β-sandwich with convex and concave regions of the two antiparallel β-sheets (Figure 2). The catalytic trio, formed by three Glu85, Glu89 and Asp87 residues, is located around the middle of the convex region of the structure (Figure 2A; magenta cpk sticks). The elongated C-terminus is folded in an α-helix and a β-strand on the concave side of the molecule and stabilised by a disulfide bridge. The PttXET16A structure is also glycosylated at Asn93 with two N-acetylglucosaminyl and one mannosyl residues that are further stabilised by the hydrogen bonds. No other atomic XET structures have been reported; however, structural features of the 3D models of nasturtium, barley (Figure 2B, Figure 3 and Figure 4), and other XETs were defined [64,66,68,69,79,81]. Although, caution should be exercised when extracting and interpreting structural information based on 3D models, where side-chain placements are known to be unreliable even at high similarities between templates and target sequences [82].

Additional information on the PttXET16A enzyme was obtained through molecular dynamics (MD) simulations [60], where the interactions between residues of the PttXET16A and two XLLG nonasaccharides revealed that one XLLG occupied the donor site and created a stable intermediate with the enzyme, while the second XLLG remained bound at the acceptor site. Notably, the reducing-end glucose moiety of XLLG at the donor -1 subsite, altered its low energy ^4^C_1_ conformation into ^1^S_3_ boat.Figure 2β-Sandwich architectures and β-jelly-roll topologies of *Populus* XET16A (PDB accession 1UMZ) in complex with the XLLG acceptor substrate (**A**), and the *Tropaeolum* XET6.3 3D molecular model in complex with the XXXG donor substrate (**B**). (**A**) *Left panel*: The XLLG acceptor (cpk sticks) bound in the *Populus* XET16A structure is indicated in dashed lines at 2.6 Å to 3.5 Å separations. The catalytic trio (Glu85, Glu89 and Asp87) is shown in cpk magenta sticks. *Right panel*: Details of the XLLG binding in *Populus* XET16A; interacting residues are marked in green cpk sticks and dots. *Bottom panel*: Some of the residues of *Populues* XET16A binding XLLG (highlighted in green) are shown in the alignment by PROMALS3D [83] of the *Populus* and *Tropaeolum* XET sequences (numbering includes signal peptides). Conservation of residues on the scale 9–6 is shown on the top of the alignment in brown. (**B**) *Left panel*: The XXXG donor (cpk sticks) bound in *Tropaeolum* XET6.3 (cyan; coordinates from [66]) is indicated in dashed lines at separations between 2.3 Å and 3.0 Å. *Right panel*: Details of XXXG binding in *Tropaeolum* XET6.3; interacting residues are marked in cyan cpk sticks in dots. *Bottom panel*: Some of the residues of *Tropaeolum* XET6.3 binding XLLG (highlighted in cyan) are shown in the alignment by PROMALS3D [83] of *Populus* XET16A and *Tropaeolum* XET6.3 sequences (numbering includes signal peptides). Conservation of residues on the scale 9–6 is shown on the top of the alignment in brown. Images were generated in the PyMOL Molecular Graphics System v2.5.2 (Schrődinger LLC, Portland, OR, USA).
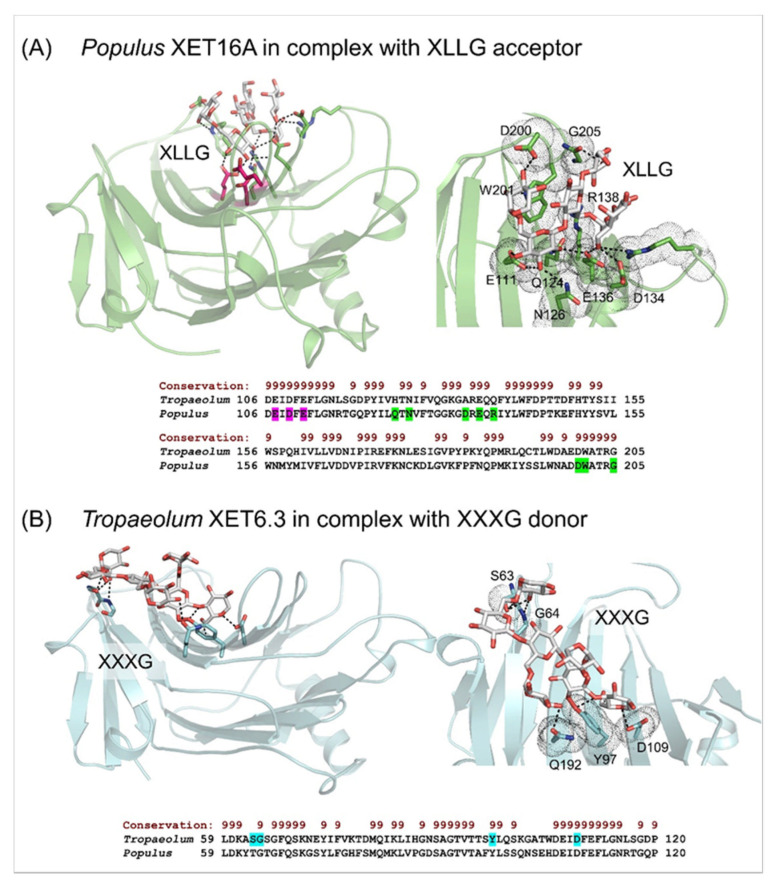


Structural differences, which determine whether the XET/XEH enzymes transglycosylate (PDB accessions 1UN1, 1UMZ) or act as the hydrolases (TmNXG1; PDB accession 2UWA) [71,80,84], were reported including the TmNXG1-DELTAYNIIG mutant (PDB accession 2VH9). Slight structural differences, which led to the stronger binding of a donor substrate combined with larger loop flexibility at the acceptor site and the flexibility of residues, underlined the hydrolytic preference [80,84].

As detailed above, structural differences modulate the binding of donor substrates combined with the loop and residues flexibilities and underline the transfer or hydrolytic preferences of XTH enzymes. This is supported by phylogenomic analyses, where XTHs segregate in the three groups, with XEHs belonging to XTH III clade, while XETs clustered within XTH I and XTH II clades [64,66,80,85]. Based on these analyses, it was concluded that GH16 hydrolases including XEHs have evolved from XETs [60,84,86].

As previously summarised [69], XET activity assays use unlabelled donor substrates and radio-chemically or fluorescently labelled acceptors, or radio-labelled donor and unlabelled acceptors. The reaction products in both cases encompass labelled saccharides, while unincorporated donors or acceptors are removed via chromatography [47,49,66], or through a gel [63,87], washed off from a filter paper [88] or ethanol precipitated [89]. When selecting an appropriate approach, it is critical to select the right type of fluorescent tag [90,91] and how to remove efficiently the surplus of substrates [48,69]. Labelled oligosaccharide probes could be also utilised in high-throughput polysaccharide microarrays [92] and through in vivo visualisation [93,94,95,96]. Supplementary XET activity assays include colorimetry [97] that relies on the formation of the blue-green-colored iodine–XG complex, and viscometry that records viscosity changes in substrates. Figure 3Molecular models of the β-sandwich architecture with the β-jelly-roll topology of *Hordeum* XET3 (top left; yellow), *Hordeum* XET4 (top right; orange), *Hordeum* XET5 (bottom left; pink), and *Hordeum* XET6 (bottom right; lemon) in complex with the XLLG acceptors (cpk sticks). Coordinates of XET3, XET4 and XET6 with XLLG were taken from [81] and those of XET5 from [64]. XLLG was docked in XET5 by HDOCK that performs docking based on a hybrid algorithm of template-based modelling and ab initio free docking [98]; the top docking pose is shown for XET5 from 100 poses ranked by energy docking scores. Details of binding of XLLG by *Hordeum* XETs are shown in dashed lines. Separations are for XET3: 2.7 Å to 3.6 Å; XET4: 2.6 Å to 3.5 Å; XET5: 2.5 Å to 3.6 Å; XET6: 2.5 Å to 3.2 Å. Images were generated in PyMOL as referenced in Figure 2.
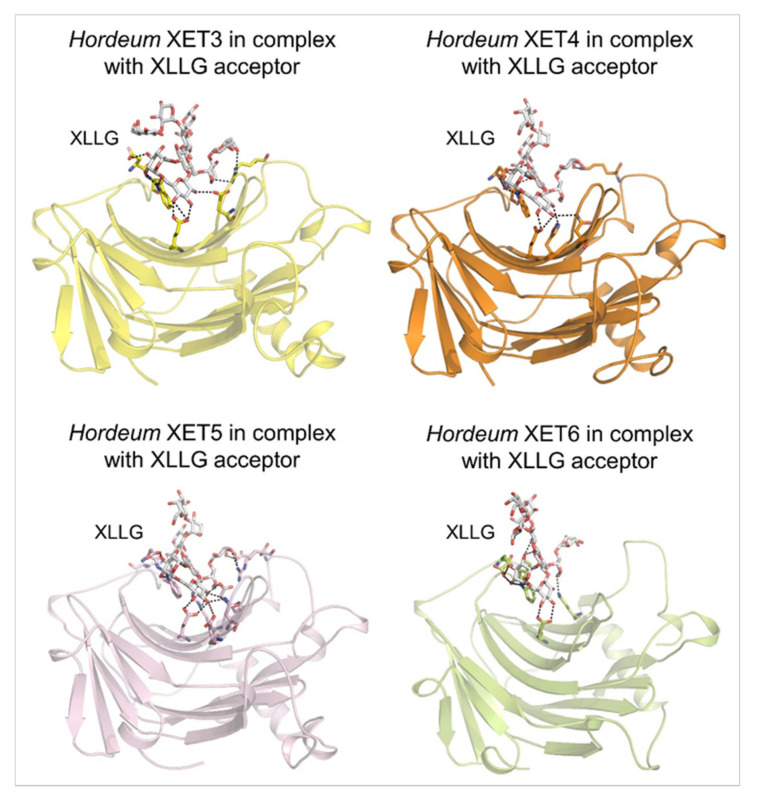


*(d) Substrate specificity of transfer reactions with xyloglucan and other than xyloglucan-derived donors and acceptors*—Substrate specificity of XETs has recently been evaluated in detail ([69]; cf. Table 1, *vide infra*); thus, here we only succinctly recapitulate key aspects of this topic.

The systematic description of xyloglucan:xyloglucosyl transferases (EC 2.4.1.207) has from the onset of their discoveries stated that these enzymes recognise XG donor (xyloglucan) and acceptor (xyloglucosyl) substrates—this catalytic activity represents (i) homo-transglycosylation reactions. Yet, since 2007 [47], the novel types of transfer reactions were described in barley XETs and later in other plant sources, termed (ii) hetero-transglycosylation reactions [63,66]. These reactions utilise the neutral donor and neutral or ionic acceptor substrates other than XG-derived [69].

*(i) Homo-transglycosylation reactions with XG-derived donors and acceptors—*The poplar PttXET16A [71,80] and *Pinus radiata* PrXTH1 [99] XETs, belonging to the I/II cluster of *XTH* gene products, were characterised as the homo-transglycosylation-type enzymes with a narrow substrate specificity that utilised XG-derived substrates. Hetero-transglycosylation activities in PttXET16A were undetected with cello-oligosaccharides [60], although the data with PrXTH1 suggested the weak binding of cellooctaose [99]. Phylogenomic analyses of the GH16 family, spanning monocots, eudicots and a basal Angiosperm [68] revealed that PttXET16A clustered with barley (*Hordeum vulgare* L.) XET5 that catalysed the reactions with XG or hydroxyethyl cellulose and XG- or cellulose-derived oligosaccharides [47].

*(ii) Hetero-transglycosylation reactions with neutral donor and neutral or ionic acceptors,**other than XG-derived—*The key question to answer regarding hetero-transglycosylation reactions is, what is the limit for the activity ratio of the XG/XG-oligosaccharide substrate pair and chemically different substrates? Another key point is that the substrate specificity of only a few XETs in near-homogenous forms has been defined, and thus this property cannot be unequivocally assigned to many XETs.

Although the broad XET substrate specificity in *Poaceae* was predicted based on molecular modelling [100], the first experimental indication that XETs could catalyse hetero-transglycosylation reactions, i.e., to mediate transfers between saccharides other than XG-derived, was presented by Ait Mohand and Farkaš [63]. This study described glycosyl transfers from XG to cello- and laminari-oligosaccharides, and from carboxymethyl and hydroxyethyl cellulose derivatives to XG oligosaccharides, using the crude protein extracts prepared from germinating nasturtium seeds [63]. This study was followed with the barley XET5 isoform [47], the first XET in a near-homogenous form, with a defined primary structure, that catalysed hetero-transglycosylation reactions in vitro. Except for XGs, this enzyme linked covalently carboxymethyl and hydroxyethyl celluloses, and (1,3;1,4)-β-d-glucans as donors and XG- and cellulose-derived oligosaccharide acceptors. The 44% efficiency of barley XET5 with the hydroxyethyl cellulose donor and the XG-oligosaccharides was comparable to that of XG, but it was significantly lower with (1,3;1,4)-β-d-glucan. Subsequently, the hetero-transglycosylation reactions were defined for a near-homogenous barley XET6 isoform [64].

Other XET enzymes recognising cellulose as the donor substrate, were partially purified XET from parsley roots (*Petroselinum crispum* Mill. Fus) [89] and near-homogenous XTH3 from *Arabidopsis thaliana* L. Heynh, which catalysed the hetero-transglycosylation reactions between cellulose and cello-oligosaccharides or cellulose and XG-oligosaccharides [65]. Despite differences in the specificity of hetero-transglycosylation reactions, barley XET5 [47] and *Arabidopsis* XTH3 [65] clustered in the same phylogenetic group of XTH I, as presumably XG-specific PttXET16A and PrXTH1 [80,99], although they segregated to different sub-clades [69]. The next XETs with known primary structures and utilising besides XG, cellulose or (1,3;1,4)-β-d-glucans as donors, were three acidic EfXTH-A, EfXTH-H and EfXTH-I isoforms from *Equisetum fluviatile* L. [101], although the homogeneity of these enzymes was not shown. While EfXTH-A displayed the comparable transfer of (1,3;1,4)-β-d-glucan fragments to XG-oligosaccharides, as was the case of barley XET5 (0.2–0.3%) [47], the activity with cellulose was higher with barley XET5. EfXTH-H and EfXTH-I showed the equivalent hetero-transglycosylation activities with (1,3;1,4)-β-d-glucans and cellulose donors [101]. Conversely, HTG and MLG (1,3;1,4-β-d-glucan): xyloglucan endotransglycosylase enzymes from *Equisetum* preferred cellulose and (1,3;1,4)-β-d-glucans with XG-oligosaccharides as respective donors and acceptors [101]. Such enzymes were so far found only in *Equisetum* and the charophytic algae [74,102], where they are predicted to re-model hemicelluloses in horsetails shoots [78]. Molecular modelling of HTG suggested the amino acid residues responsible for the evolution of their substrate specificity [79]. Figure 4Details of the XXXG donor (cpk sticks) and the [α(1-4)GalA*p*]_5_ acceptor (magenta cpk sticks) binding substrates in *Hordeum* XET3 (**A**) and *Hordeum* XET4 (**B**). In both panels, blue and black dashed lines indicate residue separations between donors or acceptors that are 2.6 Å–3.6 Å (XET3) and 2.7 Å–3.4 Å (XET4). Interacting residues are shown in yellow (XET3) and orange (XET4) cpk sticks and emphasised in dots. Subsites at -4 to -1 for XXXG and +1 to +5 for [α(1-4)GalA*p*]_5_ are indicated. Images were generated in PyMOL as referenced in Figure 2. *Bottom panel*: Some of the residues of *Hordeum* XETs that bind [α(1-4)GalA*p*]_5_ (highlighted in yellow) are shown in the alignment by PROMALS3D [83] of the *Hordeum* XET sequences (numbering includes signal peptides). Conservation of residues on the scale 9–6 is shown on the top of the alignment in brown.
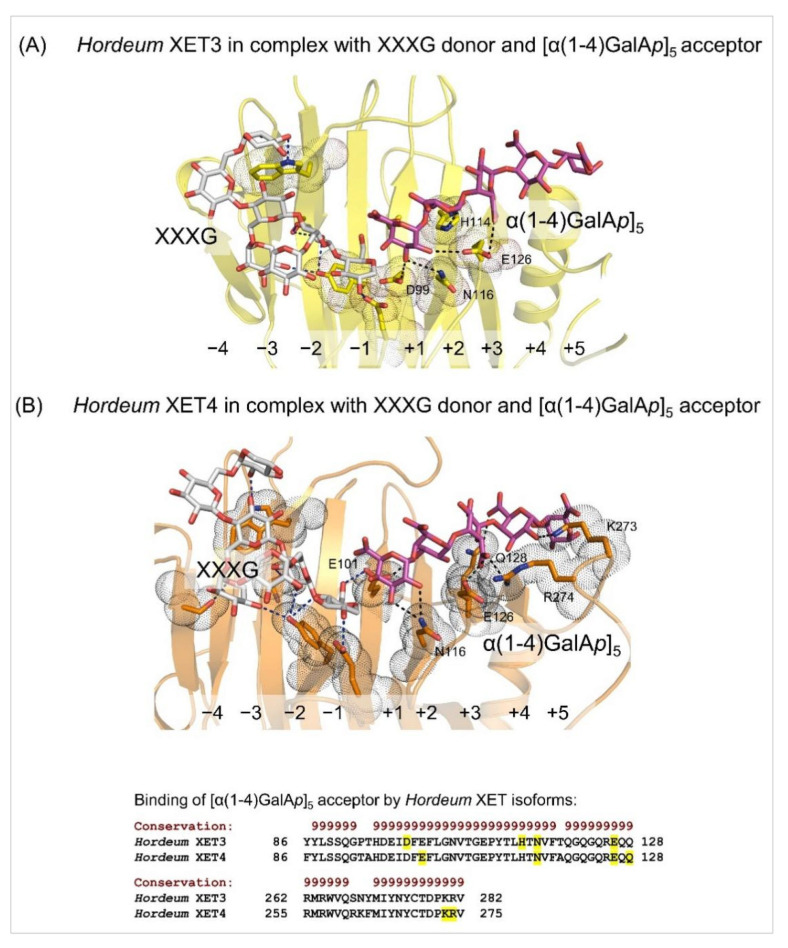



Recently, *Tropaeolum majus* L. XET6.3 classified in the XTH II clade [69] in its near-homogenous form displayed hetero-transglycosylation activity with the donor XG or hydroxyethyl cellulose and neutral oligosaccharide acceptors, such as those derived from cellulose, (1,3;1,4)-β-d-glucan, laminarin, pustulan, arabinoxylan, xylan, arabinan, arabinogalactan, mannan, glucomannan and galactomannan [66]. Figure 5Docking of the XXXG donor (cpk cyan sticks) and the xylotetraose (Xyl-OS4) acceptor (cpk magenta sticks) substrates in the active sites of *Populus* XET16A (PDB accession 1UN1) (**A**) and *Tropaeolum* XET6.3 [66] (**B**), and MD simulations of enzyme/substrate complexes after 0, 50, and 1000 ns (simulation times indicated in bottom-left corners). MD simulations were carried out similarly as described in [103]. Separations between the donors and acceptor substrates are between 2.6 Å and 3.6 Å for *Populus* XET16A and 2.7 Å–3.4 Å for *Tropaeolum* XET6.3. Residues (numbering includes signal peptides) mediating contacts with substrates are shown in cpk sticks. In *Tropaeolum* XET6.3, E136 and Q138 stabilise the binding of the Xyl-OS4 acceptor. For clarity, the residues and the subsite binding sites (−4 to −1 for XXXG and +1 to +4 for Xyl-OS) are shown in left panels only. Images were generated in PyMOL as referenced in Figure 2.
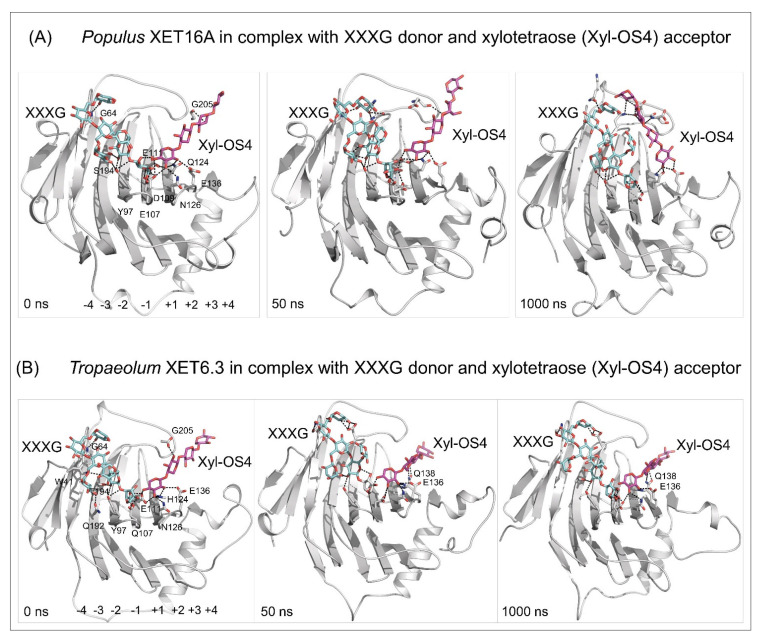



Finally, it is important to describe hetero-transglycosylation reactions that accept charged acceptors that were shown in several near-homogenous barley XETs [68] (Figure 4), where all tested neutral oligosaccharides served as the acceptors with barley XET3-XET6 isoforms, although with different efficiency. Barley XET3 and XET4 also transferred efficiently the fragments of XG or hydroxyethyl cellulose to the penta-galacturonic acid ([α(1-4)GalA*p*]_5_), a fragment of the linear part of pectin, [68]. These hetero-transglycosylation activities by barley XETs were demonstrated in vitro in CWs of barley roots using fluorescently labelled [α(1-4)GalA*p*]_5_) [68]. These findings were supported by the structural modelling of barley XET3 and XET4 with the docked XG heptasaccharide (XXXG) donor and the [α(1-4)GalA*p*]_5_ acceptor substrates in the −4 to +5 active site subsites (Figure 4), and suggested that the XET substrate poly-specificity resulted from protein sequences alterations that evoked structural re-arrangements [66,67,68,69,104].

In summary, only specific combinations of residues could underlie certain substrate specificity in XETs, which underpins the functionalisation of substrates other than XG-derived. Despite categorising these residues [66,68], the future understanding of the mechanisms of these reactions depends on the resolution of atomic structures of XETs. Future research will also be focused on in vitro and in silico studies to dissect the molecular mechanisms that drive the transglycosylation reactions catalysed by XETs (Figure 5).

## 4. Plant Cell Wall Modifications and Re-Modelling, and Methods of Investigation

*(a) Plant cell wall function, dynamics and methods of investigation—*Polysaccharide-rich nature is a key characteristic of plant CWs that through evolution reflected on a multitude of roles. These include mechanical support to counter osmotically driven turgor pressure or in the maintenance of cellular structural integrity by resisting internal hydrostatic pressures, regulation of plant growth, providing flexibility to support cell division and expansion during tissue differentiation, acting as an environmental barrier that protects cells through defence via metabolites to counteract biotic and abiotic (environmental) stresses, and for providing a milieu that through diffusion ensures cell-to-cell communication [1,2,3,4,69,105]. The evolution of changes in the plant CWs during these functions is characteristic for individual species and relies on the process of natural selection. It is exactly the pressure from surrounding environments, microorganisms, and other life forms that confront plants, which in return form CW structures that reflect and embody plant responses. This pressure, specifically from microorganisms to use plant CWs as an energy source to survive, simultaneously allows the evolution of novel biocatalysts cleaving most of the covalent linkages contained in CWs.

These multifunctional roles that underscored the evolution of plant CWs, require the formation of highly complex bio-composites, which underlines the economic feasibility of CW bioconversion to biofuels, biochemicals and other high-value products. The success of utilising the plant CWs, including the plant biomass, relies on the knowledge of the composition, structure, biosynthesis and biodegradation of CWs. Although this knowledge is not completely accessible, the structure–function relationships of CWs have become more tractable in recent years. Here, an astonishing increase in the accumulation of knowledge has occurred in the last twenty years to understand features that underlie the complexity and dynamics of plant CWs. This is fundamentally due to the progress in the physico-chemical and analytical methods, and the biological and biochemical technologies at multi-disciplinary levels, that allowed advancements to examine CWs.

These applications include the use of atomic force and scanning probe microscopy [106,107] to characterise the plant CW structure. This allows unique insights into the morphological features and molecular characteristics, such as heights, width, contour length, end-to-end, distance, and polydispersity of CW components. Other promising avenues include looking into the molecular motions and assembly formations and visualising the conformation behaviour under various conditions [107]. This technique was applied to image cellulose micro-fibrils up to a single molecular level under the near-native conditions, although other polysaccharides, namely, pectin, xanthan, carrageenan, curdlan, scleroglucan, XG, arabinoxylan and their detailed nanostructures were also characterised [107].

On the other hand, Raman spectroscopy and surface-enhanced Raman spectroscopic chemical imaging allowed investigating the mutual distribution and co-localisation of CW polysaccharides, such as pectin and XGs in, for example, onion cells [108]. Further, Felhofer and co-workers [109] described how CW thickening indentations were formed in primary CWs in algae at exactly defined areas, specifically where pectins and XGs co-localised, while in the secondary CWs, crystalline cellulose dominated in thickening areas. These observations allowed concluding that a denser network of cellulose micro-fibrils stiffened CWs at the indent and modified the CW extensibility.

Additional experimental approaches included applying one or two-dimensional Nuclear Magnetic Resonance spectroscopy to directly observe the random coil and flexible conformations of XGs in plant seeds [110], solid-state NMR to assess and quantify chemical effects (such as pH) on plant CWs [111], and confocal microscopy demonstrating that XG and cellulose formed the molecular cross-bridges connecting root border cells in pea [112]. The combination of these approaches and the equilibrium adsorptions of hemicelluloses offered new ideas, for example, that XGs could be involved in the lignification processes in primary wood CWs [113].

In conjunction with the experimental avenues, it is vital to mention theoretical MD calculations [114], based on the existing chemical and biological knowledge of plant CWs, where mutual interactions could be studied and, thus, provide the working hypotheses on how individual polysaccharides interact in CWs. For example, Zhang and co-workers [115] reported that in the water environments, van der Waals interactions prevail over electrostatic ones during the adsorption of XGs onto the Iβ 1–10 cellulose surface, and that variation in one side chain of XG had no influence on the interaction energy and the binding affinity between XGs and cellulose. Similar data provided the nano-scale views of polysaccharide interactions and showed that from five polymers (galacto-glucomannan, acetyl-galacto-glucomannan, fuco-galacto-XG, methylglucurono-xylan, methyl-glucurono-arabinoxylan), glucurono-arabinoxylan had the highest free energy of binding to cellulose nanocrystals, whereas XGs had the lowest energies amongst five models [116]. It was established that the lack of forming these interactions occurred at the inter-molecular level of hemicelluloses, rather than at the interface of cellulose, and that water molecules had a weakening effect on the hemicelluloses–cellulose interactions [115,116]—these findings might have implications for the roles of XETs in CW dynamics. Finally, a steered MD model using pull-out simulations (where force is applied to molecules along a chosen direction to understand interactions) provided the quantitative information on forces between cellulose micro-fibril surfaces [117]. These strategies are central to the deeper understanding of plant CWs, and collectively emphasise that with the mathematical modelling [118], it is vital at a microscale to tackle the challenges of understanding CW mechanics, plant tip growth, morphogenesis and stress feedback. Looking towards the future, one would expect burgeoning advances in this field because CWs are essential and indispensable to the life of plants.

*(b) Roles of xyloglucan:xyloglucosyl transferases in cell wall re-modelling*—The fundamental roles of XET enzymes in plants are in the assembly of the CW structures and in modifications during plant growth and development, in adaptations to living environments, and during biotic and abiotic stresses. This also underpins the major role of XET enzymes in the CW assembly and thickening, and in the regulation of its mechanical properties, through affecting the load-bearing framework of the XG-cellulose composites. This function of XET (together with XEH) enzymes is believed to regulate the CW plasticity along with non-catalytic expansins and other matrix polysaccharide-modifying enzymes. However, the precise molecular mechanisms of the XET activity in CWs are far from being clear [33,69]. To this end, the functional role of the specific *Arabidopsis thaliana* XET19 isoform (with undetectable hydrolase activity) was shown to increase the freezing tolerance after a low-temperature acclimation, providing a hypothesis that these enzymes modify (re-model) CWs and change their physico-chemical properties under the stress conditions [119]. Although XET enzymes have appeared concurrently with XGs in CWs of green algae (but not red or brown algae) [4,19], it remains to find out if other enzymes recognising cellulose as a donor substrate, had occurred in red or brown algae before the emergence of XET enzymes, as it is the case of HTGs [78].

The expression of *XTH* genes is regulated by hormones in response to a variety of environmental fluctuations. For this reason, XETs have evolved in multiple isoforms, which fulfil the specific roles during various stages of plant development and in response to biotic and abiotic stresses, such as drought, heat and metal toxicity [55,120,121,122,123,124,125,126,127,128]. *XTH* genes also play a unique role in the saccharide metabolism and energy storage that are connected to plant growth, as their elevated expression during elongation of cells directly affects the CW characteristics that underlie the growth and function of plant tissues through structuralisation and re-modelling [129,130].

It has now been accepted that XETs control polymerisation of plant structural polysaccharides such as XGs, and recognise cellulose [47,48,65,67,71,128,131,132,133]. XET enzymes primarily work on the XG molecules associated with cellulose micro-fibrils such that this process does not influence the integrity of CWs [1,47]. The discovery of XET isoforms that link not only XG molecules but also those of cellulose and celluose-derived oligosaccharides by barley XET5 [47] and *Arabidopsis* XTH3 [65], or XG, (1,3;1,4)-β-d-glucans and cellulose catalysed by barley XET5 [47] and *Equisetum* HTGs [67,77,78,133], or XG and a wide spectrum of structurally diverse neutral saccharides, catalysed by *Tropaeolum* XET6.3 [66], and XG and pectin fragments by barley XET isoforms [64], suggested that plants established the sophisticated and variable mechanisms to influence firmness, porosity, and flexibility (stiffness and extensibility) of CWs [47]. This allows plants to continuously modify the properties of CWs that otherwise would lack this adaptability when needed. Although it is not always known how precisely are these specific or broad specific XET isoforms expressed in plant tissues during a defined plant developmental stage, this information could be obtained from the localisation studies and gene expression analyses [66,68,77,100,134,135,136]. The significance of the transglycosylation reactions catalysed by broad specific XETs may be related to the structural roles of polysaccharides in plant CWs. These specific expression patterns thus provide a piece of evidence that it is the flexibility and properties of CWs that are required to fulfill their dynamic roles during embryogenesis, vegetative and reproductive growth, and environmental responses of plants, as these CW polysaccharides are synthesised and interact [44,45].

In conclusion, the fundamental complexities of structural polysaccharides in plant CWs have been defined [1,34,137,138,139,140], although, these complexities are not completely transparent, including the precise significance of homo- and hetero-transglycosylation reactions catalysed by XETs. Although the definition of the catalytic function of xyloglucan:xyloglucosyl transferases and more broadly of the XTH family have significantly progressed during the past fifteen years, there are limitations (and still a lack of information) as to how this knowledge can be organically implemented in the function of plant CWs. One example of how these complexities could be better tackled includes the generation of XET/XEH mutant knockout plants (in explicit plant tissues of different taxa) that would be deficient in certain XET/XEH isoforms, to understand the fine structure of plant CWs. This approach could also answer the prodigious question of the potential interchangeability in hemicellulose polymers [8], and if additional mutations to certain hemicellulose biosynthetic enzymes would activate compensatory mechanisms, while other mutations would activate feedback processes between biosynthetic pathways and endomembrane compartments [141]. The latter notions combined with the definitions of roles of broad specific XETs that generate novel hetero-polymeric networks could modify our views on plant CW architectures. Examining these novel relationships will spur a better understanding of the roles of XTH enzymes in plant CWs from macro to atomic levels. Thus, it is imperative to use the knowledge of XTHs in a CW context, amongst other applications, for the rational design of economically important crops using genetic engineering and more broadly bio-technology to increase stress tolerance, tackle drought and other abiotic stresses [105], and produce modified (thermo-responsive) XG hydrogels for bio-medicine [142] and wound healing [143].

## Figures and Tables

**Figure 1 ijms-23-01656-f001:**
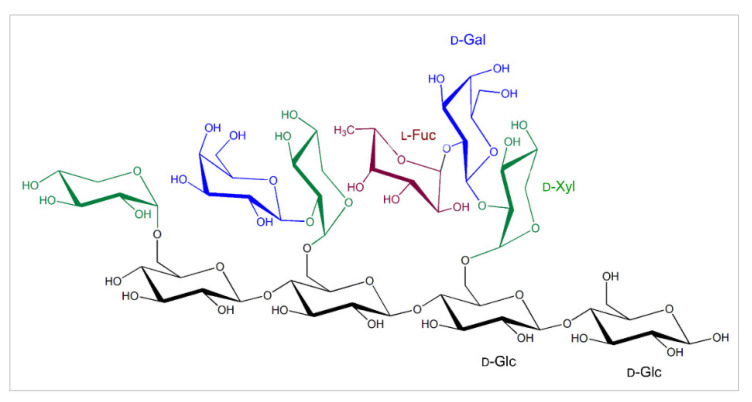
The common substitutions in the tetragluco-oligosaccharide unit of XGs that forms chains of repeating (1,4)-β-d-linked glucopyranosyl residues (black), with the C-6 carbons carrying α-d-xylopyranosyl moieties (green), which could be substituted by galactopyranosyl residues (blue) on the C-2 carbons (β-d-Galp-(1,2)-α-d-Xylp). The galactopyranosyl residues could carry fucopyranosyl residues (brown) on the C-2 carbons (α-l-Fucp-(1,2)-β-d-Galp-(1,2)-α-d-Xylp). Descriptions of sugar moieties are indicated in matching colours.

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
