# Peer review of "Broad Specific Xyloglucan:Xyloglucosyl Transferases Are Formidable Players in the Re-Modelling of Plant Cell Wall Structures"

_ijms, 2022, doi:10.3390/ijms23031656_

Round 1

Reviewer 1 Report

This manuscript examines higher plant XET enzymes and dissects how their exclusively carbohydrate-linked transglycosylation catalytic function inter-connects complex plant cell wall components and discusses progress in technologies that advance the knowledge of plant cell walls and how this knowledge defines the roles of sets as well. This manuscript is well written, I would recommend a Minor Revision before acceptance.

  • Introduction: The novelty and significance of your study should be further highlighted.
  •  Please improve the resolution of all the figures.
  • In Conclusion, they can have more discussion such as the main limitation and future works should be discussed.
  • There are some grammatical errors in this manuscript such as continuously forgetting to add ‘a’ or ‘the’ before a specific word or inappropriate use 'is' or 'are' which limits the clarity of the author’s writing.

Author Response

We thank Reviewer 1 for their thorough analyses. We have responded to each comment and introduced significant changes to the text of the manuscript as follows:

  • Introduction: The novelty and significance of your study should be further highlighted.

This section was extended (pg 1) as follows:

The additional goal of this review is to examine the roles of the xyloglucan: xyloglucosyl transferase family in plant CWs and environmental contexts, and not as isolated catalytic entities. We expect that the detailed definition of catalytic properties of these enzymes will contribute to the clarification of the biological function of xyloglucan: xyloglucosyl transferases, in the framework of the recent models of primary and secondary plant CW architectures.

  • Please improve the resolution of all the figures

The resolution was increased in all Figures 1-5.

  • In Conclusion, they can have more discussion such as the main limitation and future works should be discussed.

This section was extended (pg 13) as follows:

In conclusion, the fundamental complexities of structural polysaccharides in plant CWs have been defined [1, 34, 141-144], although, these complexities are not completely transparent, including the precise significance of homo- and hetero-transglycosylation reactions catalysed by XETs. Although the definition of the catalytic function of xyloglucan: xyloglucosyl transferases and more broadly of the XTH family have significantly progressed during the past fifteen years, there are limitations (and still a lack of information) as to how this knowledge can be organically implemented in the function of plant CWs. One example of how these complexities could be better tackled includes the generation of XET/XEH mutant knockout plants (in explicit plant tissues of different taxa) that would be deficient in certain XET/XEH isoforms, to understand the fine structure of plant CWs. This approach could also answer the prodigious question of the potential interchangeability in hemicellulose polymers [8], and if additional mutations to certain hemicellulose biosynthetic enzymes would activate compensatory mechanisms, while other mutations would activate feedback processes between biosynthetic pathways and endomembrane compartments [145]. The latter notions combined with the definitions of roles of broad specific XETs that generate novel hetero-polymeric networks could modify our views on plant CW architectures. Examining these novel relationships will spur a better understanding of the roles of XTH enzymes in plant CWs from macro to atomic levels. Thus, it is imperative to use the knowledge of XTHs in a CW context, amongst other applications, for the rational design of economically important crops using genetic engineering and more broadly bio-technology to increase stress tolerance, tackle drought and other abiotic stresses [106], and produce modified (thermo-responsive) XG hydrogels for bio-medicine [146] and wound healing [147].

  • There are some grammatical errors in this manuscript such as continuously forgetting to add ‘a’ or ‘the’ before a specific word or inappropriate use 'is' or 'are' which limits the clarity of the author’s writing.

We have revised the text of the manuscript (grammar and syntax) and corrected it. 

Reviewer 2 Report

That xyloglucan endo-transglycosylases are important for remodeling plant cell walls is not remarkable or particularly noteworthy.  Presenting descriptive narratives of biochemical activities, molecular structures and analytical procedures and processes does not make for an interesting review article.  Although the article is informative in several aspects related to plant molecular structures and techniques, it fails to present clear, definitive evidence pertaining to the relevance of xyloglucosyl transferases in cell wall remodeling and associated activities.

I was struck by the authors’ closing statement:

“In conclusion, it is imperative to use the knowledge of XTHs in a CW context, amongst other applications, for the rational design of economically important crops using genetic engineering and more broadly biotechnology to increase stress tolerance, tackle climate change, and produce modified (thermo-responsive) XG hydrogels for bio-medicine and wound healing.”

Utilizing knowledge of the biochemistry and structure of any molecule(s) of interest is standard procedure to genetically engineer a plant model system.  How can such knowledge be used to “tackle climate change”?  Can the authors define climate change?

I suggest that the authors consider a more serious intellectual approach to this article.  All the figures need to be of higher quality for publication.

Author Response

We thank Reviewer 2 for their thorough analyses. We have responded to their critical comment and introduced changes to the text of the manuscript as follows:

(i)         We agree with the Reviewer, that the statement regarding ‘climate change’ was broad and misleading. We have revised the entire paragraph on page 13, which was rephrased as follows:

In conclusion, the fundamental complexities of structural polysaccharides in plant CWs have been defined [1, 34, 141-144], although, these complexities are not completely transparent, including the precise significance of homo- and hetero-transglycosylation reactions catalysed by XETs. Although the definition of the catalytic function of xyloglucan: xyloglucosyl transferases and more broadly of the XTH family have significantly progressed during the past fifteen years, there are limitations (and still a lack of information) as to how this knowledge can be organically implemented in the function of plant CWs. One example of how these complexities could be better tackled includes the generation of XET/XEH mutant knockout plants (in explicit plant tissues of different taxa) that would be deficient in certain XET/XEH isoforms, to understand the fine structure of plant CWs. This approach could also answer the prodigious question of the potential interchangeability in hemicellulose polymers [8], and if additional mutations to certain hemicellulose biosynthetic enzymes would activate compensatory mechanisms, while other mutations would activate feedback processes between biosynthetic pathways and endomembrane compartments [145]. The latter notions combined with the definitions of roles of broad specific XETs that generate novel hetero-polymeric networks could modify our views on plant CW architectures. Examining these novel relationships will spur a better understanding of the roles of XTH enzymes in plant CWs from macro to atomic levels.

Thus, it is imperative to use the knowledge of XTHs in a CW context, amongst other applications, for the rational design of economically important crops using genetic engineering and more broadly bio-technology to increase stress tolerance, tackle drought and other abiotic stresses [106], and produce modified (thermo-responsive) XG hydrogels for bio-medicine [146] and wound healing [147].

(ii)        All the figures need to be of higher quality for publication.

The resolution was increased in all Figures 1-5.

(iii)       (x) Moderate English changes required

We have revised the text of the manuscript (grammar and syntax) and corrected it.

Round 2

Reviewer 2 Report

Much better.